# Assessment of STAT4 Variants and Risk of Hepatocellular Carcinoma in Latin Americans and Europeans

**DOI:** 10.3390/cancers15184530

**Published:** 2023-09-12

**Authors:** Alan Ayoub, Chimaobi M. Anugwom, Jhon Prieto, Domingo Balderramo, Javier Diaz Ferrer, Angelo Z. Mattos, Marco Arrese, Enrique Carrera, Zwier M. A. Groothuismink, Jeffrey Oliveira, Andre Boonstra, Jose D. Debes

**Affiliations:** 1Faculty of Medicine, University of Zagreb, 10000 Zagreb, Croatia; alan.ayoub555@gmail.com; 2Department of Medicine, Division of Gastroenterology, Division of Infectious Disease, University of Minnesota, Minneapolis, MN 55455, USA; anugw001@umn.edu; 3Health Partners Digestive Care, Saint Paul, MN 55130, USA; 4Centro de Enfermedades Hepaticas y Digestives, Bogota 110121, Colombia; jhonprieto123@yahoo.com; 5Department of Gastroenterology, Hospital Privado Universitario de Córdoba, Instituto Universitario de Ciencias Biomédicas de Córdoba, Córdoba X5016, Argentina; dcbalderramo@yahoo.com; 6Department of Gastroenterology, Universidad San Martin de Porres, Lima 15024, Peru; jodf13@hotmail.com; 7Graduate Program in Medicine: Hepatology, Federal University of Health Sciences of Porto Alegre, Porto Alegre 90050-170, Brazil; angmattos@hotmail.com; 8Department of Gastroenterology, Pontificia Universidad Católica de Chile, Santiago 3580000, Chile; marrese@med.puc.cl; 9Department of Gastroenterology, Universidad San Francisco de Quito, Quito 170901, Ecuador; carreraestupinan.enrique@gmail.com; 10Department of Gastroenterology and Hepatology, Erasmus Medical Center, 3015 CE Rotterdam, The Netherlands; z.groothuismink@erasmusmc.nl (Z.M.A.G.); j.oliveira@erasmusmc.nl (J.O.); p.a.boonstra@erasmusmc.nl (A.B.)

**Keywords:** hepatocellular carcinoma, single nucleotide polymorphism, *STAT4*

## Abstract

**Simple Summary:**

Hepatocellular carcinoma (HCC) is a prevalent and fatal type of liver cancer with various risk factors. This study examines the connection between a specific genetic variant, *STAT4* rs7574865, and HCC risk in Latin American and European populations. The results reveal no general association between this genetic variant and HCC in the studied groups. This study underscores the significance of researching diverse populations to gain a better understanding of the broader influence of genetic factors on HCC risk, which may aid in developing more effective strategies for identifying and managing high-risk individuals.

**Abstract:**

Hepatocellular carcinoma (HCC) is the third leading cause of cancer death worldwide. The *STAT4* rs7574865 genetic variant has been associated with an increased risk of developing HCC in Asian populations. However, this association has not been studied in Latin America and is poorly assessed in European populations. This case-control study investigated the association between *STAT4* rs7574865 and HCC risk in these populations. We evaluated DNA samples from seven medical institutions across six Latin American countries and one Dutch institution in 1060 individuals (344 HCC and 716 controls). *STAT4* rs7574865 SNP was genotyped using TaqMan-genotyping assay and analyzed using logistic regression. We found no significant association between the homozygous risk allele (G) of *STAT4* and HCC development in either population, with odds ratios (OR) for GG versus TT of 0.85 (CI: 0.48–1.52, *p* = 0.58) and 0.81 (CI: 0.34–1.93, *p* = 0.67) for Latin Americans and Europeans respectively. No correlation was found between the risk allele and HCC based on underlying liver disease. However, we found that Latin Americans of European ancestry were more likely to carry the risk allele. Our results suggest that the *STAT4* SNP rs7574865 does not influence the risk of developing HCC in Latin American or European populations, highlighting the importance of evaluating genetic risk factors in various ethnic groups and understanding the possible influence of ancestry on the genetic basis of disease.

## 1. Introduction

Hepatocellular carcinoma (HCC) is the most common cause of primary liver cancer and the third leading cause of cancer death globally [1]. The significant risk factors for HCC include infections with hepatitis B virus (HBV) and hepatitis C virus (HCV), as well as alcohol-associated cirrhosis and metabolic dysfunction-associated steatotic liver disease (MASLD) [2]. Both MASLD and HBV are particularly important in Latin America. The impact of MASLD has been steadily growing worldwide, given the obesity epidemic, with a particular emphasis in Latin America [3,4,5]. Current trends show that by 2040, liver cancer deaths related to MASLD among Latin Americans will increase by 85% compared to the global 59%, and recent data have shown a high rate of advanced-stage HCC diagnosis in Latin America, despite current developments in diagnostics and surveillance [6,7,8,9].

Regarding HBV, a previous study by our group in Latin America showed a significant proportion of early HCC and high mortality related to HBV infection, suggesting that HBV is an underreported cause of early HCC in the region [10]. The stage of HCC diagnosis is highly predictive of mortality, with high mortality rates associated with diagnosis of HCC in the later stages of the disease [11]. Therefore, successful identification and determination of high-risk individuals can play a crucial role in the earlier diagnosis of HCC.

HCC risk assessment proposed by genetic determination is an attractive and applicable path. HCC has a considerable genetic component, and a family history doubles the risk for HCC development [12]. Single nucleotide polymorphisms (SNPs) are common genetic variations that can significantly influence an individual’s susceptibility to various diseases [13]. Among those variations, an SNP on the gene *STAT4* rs7574865 has been increasingly implicated in the development of HCC, mainly in Asian populations. This SNP consists of two alleles, G and T, with the G allele being associated with an increased risk of developing HCC [14]. This SNP has been previously associated with autoimmune diseases such as autoimmune thyroid disease, ulcerative colitis, and malignancies such as HCC [15,16]. *STAT4* expression has also been associated with numerous malignancies, such as ovarian and lung cancer [17,18]. It has been theorized that reduced *STAT4* expression, induced by the SNP, could potentially weaken the anti-tumor response by affecting *STAT4* modulation of interleukin 12 in NK cells [19].

Building on this mechanistic understanding, numerous studies have assessed its clinical relevance, confirming a notable association between this SNP in *STAT4* and HCC in different populations. Jiang et al. found an association between *STAT4* and HCC among patients with HBV-related HCC in a genome-wide association study (GWAS) performed in China [14]. While other studies, such as the replication study by Chen et al., failed to repeat the same findings [20]. However, these studies have been performed in Asian and Caucasian populations with no representation of other ethnic groups [14,20,21,22,23]. Appendix A describes all previous studies on *STAT4* and HCC and their methodologies. To our knowledge, no evaluations of the genetic association between *STAT4* and HCC have been conducted in Latin Americans. The current study addresses this knowledge gap by focusing on a Latin American cohort. Specifically, we hypothesized that a different risk allele frequency would be found in this population due to differential genetic background.

## 2. Methods

### 2.1. Samples and Study Subjects

This is a cross-sectional study focusing on patients who were incorporated into the ESCALON network (www.escalon.eu, accessed on 1 August 2023). For this study, we included samples from seven medical institutions in six Latin American countries (Argentina, Ecuador, Brazil, Chile, Peru and Colombia) alongside a single Dutch institution that served as the source of the European subjects. Participants were chosen for the study as per continuous participation in the ESCALON study (prospective follow-up for HCC biomarkers) rather than randomly [24,25].

At each participating medical center, patient data and blood samples were gathered and recorded in a Research Electronic Data Capture (REDCap) registry. To uphold ethical standards, all patient data was anonymized, and the provision of previous informed consent (in the patient’s respective language) and ethics approval from all participating institutions was mandatory. Upon inclusion in the study, control participants are followed every six months in their respective clinics with clinical, ultrasound and laboratory evaluations. This allows the investigators to determine the lack of HCC development for the duration of the study. Participants do not receive remuneration or simile for participation as it is part of their recommended care by being at risk for HCC.

Patients were classified as having HCC if they met the American Association for the Study of Liver Disease’s prescribed biopsy or imaging standards [26]. The diagnostic process involved two main avenues: histological confirmation and imaging studies. A liver biopsy was performed for histological confirmation, which involved extracting tissue samples from the liver lesion for microscopic examination. This method was primarily reserved for cases where imaging studies yielded inconclusive results, or additional histopathological information was deemed clinically necessary. For imaging-based diagnosis, a multiphasic computed tomography (CT) or magnetic resonance imaging (MRI) was utilized. Specific criteria, such as arterial phase hyper-enhancement followed by venous phase washout, had to be met to confirm an HCC diagnosis solely on imaging grounds. Dual-phase or triple-phase techniques were employed, capturing images at multiple time intervals post-contrast administration to assess the vascular behavior of the lesion.

Patients were classified as having alcohol-related liver disease if they had persistent steatohepatitis in the setting of prolonged ethanol intake, defined as 30 g/day for women and 40 g/day for men over ten years. The diagnosis of MASLD was determined by the managing hepatologist or by evidence of hepatic steatosis on pathology or imaging in the absence of other clear reasons for hepatic steatosis. Individuals without viral hepatitis, MASLD or alcohol-related liver disease were categorized as “other” etiology, which encompassed both known causes and unknown causes. If patients were identified with a mixed etiology—having any combination of HBV, HCV, MASLD, and alcohol-related liver disease—they were excluded from the subgroup analysis.

To ensure accuracy and integrity, all collected data underwent regular audits and was independently verified by the overseeing institution before initiating data analysis.

### 2.2. Genotypic Analysis

To assess single nucleotide polymorphisms, genomic DNA was obtained from whole blood samples utilizing Gentra Puregene (Minneapolis, MN, USA), following the steps outlined in the manufacturer’s instructions. The *STAT4* SNP rs7574865 G > T was specifically genotyped using pre-configured TaqMan probe SNP genotyping assays from Thermo Fisher (cat. nr. 4351376). The context sequence of the probes is TATGAAAAGTTGGTGACCAAAATGT[G/T]AATAGTGGTTATCTTATTTCAGTGG, with the variation G versus T as indicated. The TaqMan SNP genotyping assay aims to determine genetic variations within genomic DNA by identifying a single nucleotide difference within a specific gene locus. Each TaqMan SNP Genotyping Assay includes two differentially labeled, allele-specific TaqMan MGB probes and a PCR primer pair that uniquely amplify and provide unmatched specificity for the allele of interest. Upon amplification by real-time PCR, the fluorescent signal of the probes is registered and visualized in allelic discrimination plots. The genotyping was carried out with the StepOne-Plus Real-Time PCR System, in combination with a Custom TaqMan SNP Genotyping Assay (Applied Biosystems, Waltham, MA, USA). Each quantitative PCR reaction had a 10 µL volume, comprising 4 µL of genomic DNA and a 6 µL genotyping master mix inclusive of the probe.

### 2.3. Statistical Analysis

The comparison of demographic and clinical traits across various subjects involved presenting continuous variables as means and standard deviations, while categorical variables were depicted as proportions. Chi-squared, Fisher’s Exact test and multiple logistic regression were performed to evaluate the association between this SNP and HCC. The odds ratios (OR) and 95% confidence intervals (CI) for each variable were determined. JASP (Version 0.17.1) [Computer software] was utilized for performing these statistical calculations. JASP software was chosen for our statistical analysis due to its user-friendly interface, versatile range of statistical tests, including chi-square tests and logistic regression models, and its robust performance in handling large data sets, making it particularly suitable for the study’s requirements.

### 2.4. Ethics

Each institution involved in the study provided ethical approval for the research as required for ESCALON. Before their involvement in the ESCALON study, all participants provided written informed consent in their native language.

## 3. Results

### 3.1. Patient Characteristics

We included 1060 individuals: 344 patients with HCC (155 Latin Americans, 189 Europeans) and 716 controls (454 Latin Americans, 262 Europeans). The controls consisted of patients who had liver disease but no HCC. Of the Latin American controls, 80% had cirrhosis, while 60% of the European controls had cirrhosis. The median age for Latin Americans with HCC and controls was 68 (IQR 62–73) and 63 (IQR 57–69) years, respectively, while for the European cohort, it was 67 (IQR 61–71) and 58 (IQR 46–66) years, respectively. Males represented 64% of the Latin American HCC cohort and 77% of the European HCC cohort.

The most common causes of underlying liver disease in Latin Americans were MASLD, 46% and 59% in the study group and controls, respectively, followed by alcohol use disorder (AUD), 25% and 15%, respectively. In the European cohort, AUD and MASLD predominated in the HCC group (35% and 19%, respectively), with HBV and HCV being more common among controls (31% and 29%). All patient characteristics are summarized in Table 1.

### 3.2. STAT4 HCC Risk Assessment

Latin Americans with HCC presented with lower frequencies of the risk-associated GG genotype compared to their counterparts without HCC (37% vs. 42%, respectively, Figure 1). However, this difference was not statistically significant (*p* = 0.46). Similarly, a non-significant lower frequency of the GG genotype was observed in Europeans with HCC (58%) compared to those without HCC (62%, *p* = 1). There was no significant statistical difference in the frequencies of the TT genotype and GT genotype between participants with and without HCC in either cohort (Table 2). Overall, the G allele (risk allele) did not determine risk for HCC in either cohort yet showed a trend towards a non-statistically significant reduction in risk in both groups with odds ratios (OR) for GG versus TT of 0.85 (CI: 0.47–1.52) and 0.81 (CI: 0.34–1.93) for Latin Americans and Europeans respectively. We later performed a risk assessment of the G allele between cirrhotic and non-cirrhotic patients but found no correlation among both groups in either cohort (Appendix A). Risk allele frequency was also evaluated using the gnomAD database, a genomic data aggregation platform incorporating 17,720 sequences related to Latin Americans (https://gnomad.broadinstitute.org/, accessed on 5 June 2023) [27]. The risk allele frequency within the gnomAD database registered at 64.8% for Latin Americans; in our Latin Americans HCC cohort, it was found to be 60.52%. Importantly, this did not differ significantly from the risk allele frequency in our cohort of Latin Americans with cirrhosis but without HCC (OR = 0.92, 95% CI: 0.70–1.20, *p* = 0.53).

### 3.3. STAT4 HCC Risk Assessment Based on Underlying Liver Disease

We performed subgroup analyses focused on underlying liver disease, but no association was found between the GG genotype and HCC in Latin Americans diagnosed with MASLD (GG vs. GT and TT OR = 0.78, 95% CI: 0.43–1.38, *p* = 0.414). We also did not find significant correlations between the GG genotype and HCC in Latin Americans with viral hepatitis or alcohol-related liver disease (Table 3). A specific assessment was performed on those with HBV. However, the number of HBV samples was too low to perform a reasonable analysis (Table 3). Similarly, for the European cohort, no correlation was observed between the GG genotype and HCC in subgroup analyses based on the type of underlying liver disease (Table 3).

### 3.4. Effect of Ancestry in HCC Risk Related to STAT4

Most (87%) Latin Americans diagnosed with HCC or cirrhosis examined in this study had non-European ancestry. The risk allele frequency for Latin Americans of non-European descent was 60.9%, whereas it was 69.38% for European descent (*p* = 0.04). This suggests that Latin Americans of European ancestry are more likely to carry the G allele (OR = 1.46, 95% CI: 1.02–2.08, Figure 2A). Crucially, there was no significant difference in risk allele frequency between Europeans and Latin Americans of European ancestry (*p* = 0.07). The same applied when comparing Europeans with HCC to Latin Americans of European ancestry with HCC (*p* = 0.76) (Figure 2B). Another analysis showed that Latin Americans with HCC are significantly less likely to carry the GG genotype than Europeans with HCC (OR = 0.424, 95% CI: 0.27–0.66, *p* < 0.001). However, this significance was not seen when observing patients of European descent (OR = 1.4, 95% CI:0.46–4.29, *p* = 0.78) (Figure 3).

## 4. Discussion

Our analysis found no statistically significant association between the *STAT4* rs7574865 G > T SNP and HCC in a Latin American population. To our knowledge, this is the first study on the effect of the *STAT4* SNP on HCC development in patients from Latin America. In addition, we evaluated a cohort from Europe for comparative purposes, and consistent with other prior findings, we discerned no correlation between the SNP and the progression of HCC in patients with liver disease [20,28,29,30]. This study emphasizes the need to entertain population background in the clinic when addressing genetic risk for HCC. Moreover, it suggests that future research should focus on multiple and broad backgrounds when assessing a specific biomarker (be genetic, immune, or other) to risk-stratify the approach to HCC.

*STAT4* functions as a mediator of immunity and tumor growth as an important member of the JAK-STAT pathway. It has been associated with autoimmune diseases [31,32] and malignancies such as lung, colon, and breast cancer [22,23,33,34,35,36,37].

A mechanism proposed by El Sharkawy et al. suggested that the G risk allele is linked with reduced *STAT4* mRNA expression, subsequently impairing interleukin 12 signaling in NK cells, thereby weakening their anti-tumor efficacy [22]. However, our study found no differences between cirrhotic and non-cirrhotic populations in terms of *STAT4* risk allele. Our study found that *STAT4* mutation does not seem to be associated with HCC in our populations (both Latin American and European). However, future studies demonstrating the functional consequences of the SNP are needed to determine whether the polymorphism affects the expression levels of *STAT4* in selected immune cells and whether the phosphorylation of *STAT4* is affected upon stimulation of these cells with known stimulators such as interleukin-12.

The discrepancy in our results compared to existing literature may not only stem from the ethnic diversity of our cohort compared to the vastly homogenous Chinese population but also the underlying pathophysiological mechanisms of HCC development [38,39,40,41,42,43]. For instance, the role of *STAT4* in HCC development may differ depending on the etiology of liver disease. While hepatitis B virus (HBV) is the primary cause of HCC in Chinese populations [42], only 9% of HCC cases in our study were HBV-related, potentially affecting the impact of *STAT4* polymorphisms on disease progression. Indeed, Zhong et al. had 74% of HBV-related HCC, and Chen et al. had 82% of HBV-related HCC in their studies [44,45]. This difference in the underlying cause of HCC reflects the different mechanisms of HCC development and the role of *STAT4* in its occurrence in different etiologies of liver disease. Due to a low number of samples, we could not assess the impact of the *STAT4* SNP in those exclusively with HBV-related HCC.

El Sharkawy et al. have recently confirmed that a risk *STAT4* SNP in a Caucasian population is associated with a greater rate of fibrosis and inflammation in HBV patients [22]. The same effects have been described in Asian populations [23,34,36,37]. It is also possible that inflammatory mechanisms leading to different immune active mechanisms related to STAT activation or inactivation are present in certain populations compared to others, affecting HCC risk [20,46,47]. This could be associated with a multiple-hit development with environmental exposures (such as aflatoxins) and specific SNPs, eventually leading to HCC development [44].

Importantly, our study also evaluated self-reported heritage background. Latin Americans of European descent showed a similar distribution of the risk allele compared to the European cohort in the overall population (partially supporting self-reported heritage as a reasonable factor when full genome analysis is unavailable) and no difference in SNP prevalence among HCC or controls. It is important to note that our study relied on self-reported heritage background without any genetic analysis, which could introduce bias. Although this approach enhances the data and provides preliminary findings related to ancestry, this association should be confirmed in the setting of admixture studies [41]. Furthermore, the sample size in our study was relatively small, limiting the generalizability of our findings. Therefore, larger studies are needed to confirm and further investigate these associations.

The current findings do not yet inform clinical practice in terms of better diagnosis or treatment of HCC. However, understanding the value of *STAT4* for risk-stratification in HCC could potentially guide future strategies. Moreover, our study warns against extrapolating the risk profile of SNPs from different populations.

## 5. Conclusions

This study provides novel evidence regarding the association of the *STAT4* rs7574865 G > T single nucleotide polymorphism and hepatocellular carcinoma (HCC) within Latin American populations. Our results indicate no significant association between this SNP and the occurrence of HCC in the studied Latin American cohorts compared to a European control group. Future research should aim to include larger sample sizes and more diverse ethnic groups to validate these findings.

## Figures and Tables

**Figure 1 cancers-15-04530-f001:**
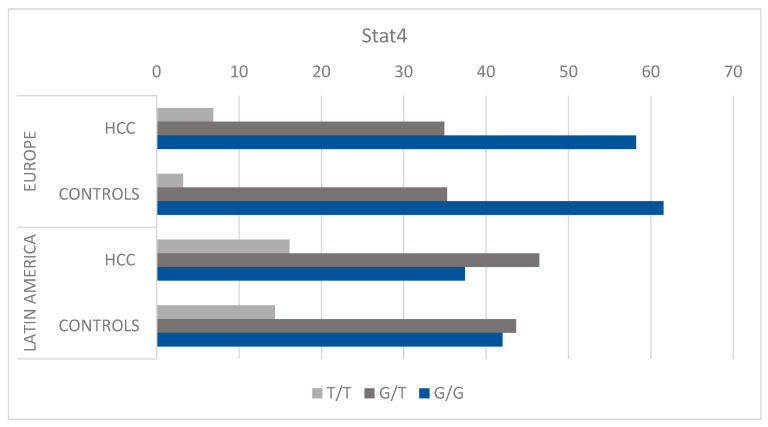
Distribution of genotypes. The genotype frequency distribution of the *STAT4* rs7574865 G > T single nucleotide polymorphism in patients with hepatocellular carcinoma compared to controls without hepatocellular carcinoma with liver disease. Genotype frequency is stratified by geographic patient location in two separate cohorts, Europeans and Latin Americans. Percentage of each genotype among the European HCC and control groups, respectively: GG (58.2%, 61.54%), GT (34.93%, 35.26%), TT (6.88%, 3.21%). Percentage of each genotype among the Latin American HCC and control groups, respectively: GG (37.42%, 41.99%), GT (46.45%, 43.65%), TT (16.13%, 14.36%). Abbreviations: HCC, hepatocellular carcinoma.

**Figure 2 cancers-15-04530-f002:**
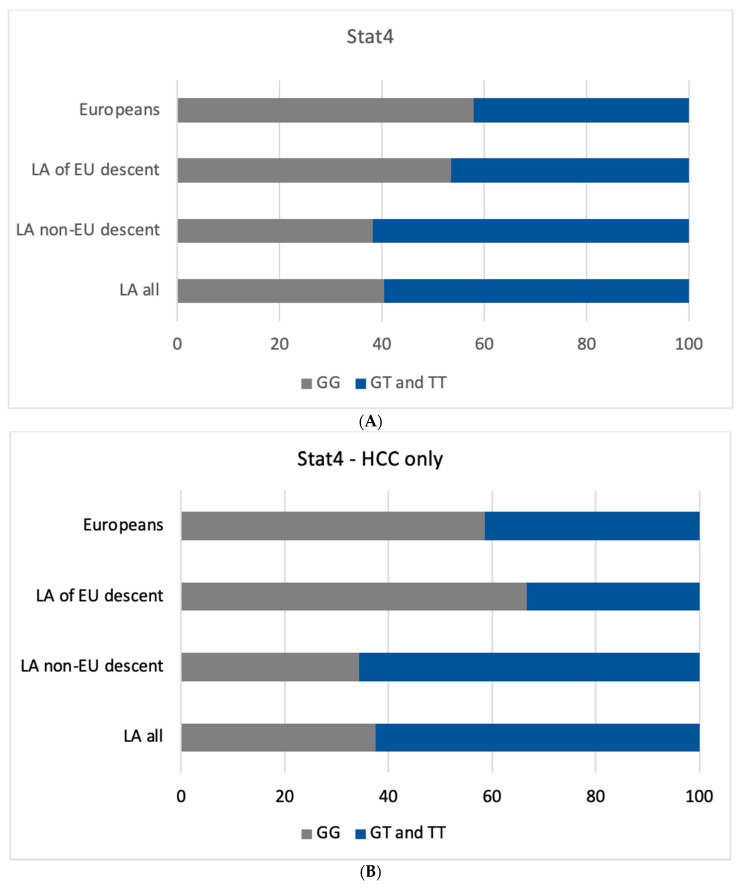
(**A**) Genotype Frequency Distribution of *STAT4* rs7574865 G > T SNP Across Different Geographic and Ethnic Cohorts. The genotype frequency distribution of the *STAT4* rs7574865 G > T single nucleotide polymorphism in all patients. Frequencies are compared between patients of different current geographic locations and descent with a European cohort, a Latin American cohort of European descent and a Latin American cohort of non-European descent. This figure shows the similarities in the distribution of the GG genotype between Europeans and Latin Americans of European descent. (**B**) Genotype Frequency Distribution of *STAT4* rs7574865 G > T SNP in HCC patients. The genotype frequency distribution of the *STAT4* rs7574865 G > T single nucleotide polymorphism in HCC patients. Frequencies are compared between patients of different current geographic locations and descent with a European cohort, a Latin American cohort of European descent and a Latin American cohort of non-European descent. This figure shows the differences in the distribution of the GG genotype between different cohorts. The *x*-axis indicates the frequency of each genotype measured in percentages, while the *y*-axis represents different patient cohorts. Each bar is color-coded to differentiate between the GG, GT, and TT genotypes. Abbreviations: LA, Latin Americans; EU, European.

**Figure 3 cancers-15-04530-f003:**
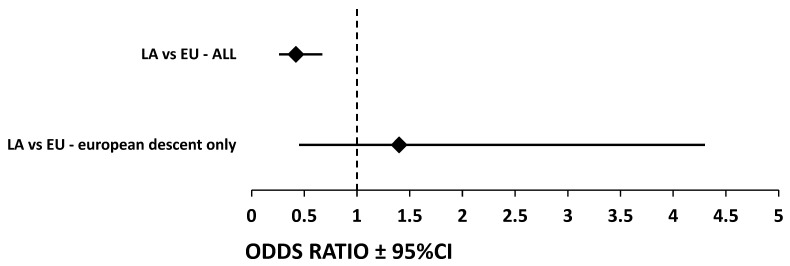
Differences in GG Genotype Prevalence Between Latin American and European Cohorts: Odds Ratios and 95% Confidence Intervals. Illustrating the Difference in GG Genotype Prevalence between Patient Cohorts. This figure visualizes the odds ratios (OR) and 95% confidence intervals (CI) for the prevalence of the GG genotype in *STAT4* rs7574865 G > T SNP across different patient groups. The *y*-axis delineates two key comparisons: the first line represents the comparison between Latin Americans and Europeans, while the second line compares Latin Americans of European descent with Europeans of European descent. The *x*-axis shows the OR values along with the corresponding 95% CIs. For the comparison between Latin Americans and Europeans, the OR is 0.424 with a 95% CI of 0.27–0.66, indicating a significant difference. Conversely, the analysis comparing Latin Americans of European descent and Europeans yielded an OR of 1.4 with a 95% CI of 0.46–4.29, suggesting a lack of significant difference. Each OR is represented by a square, and the horizontal lines extending from it illustrate the range of the 95% CI—abbreviations: LA, Latin Americans; EU, Europeans; CI, Confidence interval.

**Table 1 cancers-15-04530-t001:** Basic patient characteristics.

Characteristics	Latin America	European
HCC (*n* = 344)	*n* = 155	*n* = 189
Age [median (IQR)] years	68 (62–73)	67 (61–71)
Male, *n* (%)	99 (64%)	145 (77%)
Cause of Liver disease, *n* (%)		
Hep. B virus (HBV)	6 (4)	21 (11)
Hep. C virus (HCV)	23 (15)	23 (12)
MASLD/MASH	71 (46)	36 (19)
Alcohol	38 (25)	66 (35)
Autoimmune	4 (3)	5 (3)
Other	3 (2)	33 (17)
None	10 (6)	4 (2)
Cirrhosis (*n* =716)	*n* = 454	*n* = 262
Age [median (IQR)] years	63 (57–69)	58 (46–66)
Male, *n* (%)	220 (48%)	167 (64%)
Cause of Liver disease, *n* (%)		
Hep. B virus (HBV)	16 (4)	80 (31)
Hep. C virus (HCV)	40 (9)	76 (29)
MASLD/MASH	267 (59)	43 (16)
Alcohol	67 (15)	18 (7)
Autoimmune	25 (6)	21 (8)
Other	36 (8)	23 (9)
None	3 (1)	0

**Table 2 cancers-15-04530-t002:** Odds of HCC divided by cause of liver disease and ethnicity.

Disease	OR		CI 95%		*p*	
	LA	EU	LA	EU	LA	EU
MASLD/MASH	0.778	1	0.431–1.382	0.440–2.589	0.414	1
AIH	2.066	0.5	0.252–17.927	0.064–3.906	0.592	0.597
ALCOHOL	0.578	1.688	0.257–1.299	0.577–4.939	0.223	0.413
HBV	0.4	0.813	0.036–4.411	0.310–2.138	0.623	0.805
HCV	1.143	1.364	0.376–3.472	0.516–3.602	1	0.631

**Table 3 cancers-15-04530-t003:** Odds of HCC stratified by *STAT4* risk allele.

Latin Americans	OR [95 CI]	*p* Value
GG vs. TT	0.85 [0.48, 1.52]	0.58
GG vs. (GT and TT)	0.85 [0.58, 1.25]	0.46
(GG and GT) vs. TT	0.92 [0.55, 1.56]	0.8
**Europeans**		
GG vs. TT	0.81 [0.34, 1.93]	0.67
GG vs. (GT and TT)	1.01 [0.68, 1.5]	1
(GG and GT) vs. TT	0.79 [0.34, 1.87]	0.56

## Data Availability

The data that support the findings of this study are available on request from the corresponding author (JD).

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
