# Peer review of "Assessment of STAT4 Variants and Risk of Hepatocellular Carcinoma in Latin Americans and Europeans"

_cancers, 2023, doi:10.3390/cancers15184530_

Round 1
Reviewer 1 Report
The manuscript titled ‘’ Assessment of STAT4 variants and risk of Hepatocellular carcinoma in Latin Americans and Europeans’’ showed interesting results investigating the association between STAT4 rs7574865 and HCC risk in Latin and European populations. The study needs more improvement before being considered for publication including;
1. More introduction about the studied polymorphisms is expected. Why were these concrete polymorphisms selected? What is the proposed mechanism by which these synonymous variants may affect the risk of HCC cancer?
2. The discussion part is rather simple, mainly repeating the results, but generally lacking the real discussion of obtained results with previously published studies of association of selected polymorphisms with either HCC cancer or other oncological diseases. Mechanisms for the discrepancy data should be clarified.
3. In materials and methods, no data about the selection of rs mutation such as Assay ID,… so readers and researchers can reproduce data using the same assay rs.
4. No additional post-tests have been performed to confirm the genotyping data either on gene (mRNA) or protein level.
Author Response
We thank the reviewers for their time and thoughtful review of our manuscript. We have addressed all their comments in a detailed response below, with answers in italic.
Reviewer 1
- More introduction about the studied polymorphisms is expected. Why were these concrete polymorphisms selected? What is the proposed mechanism by which these synonymous variants may affect the risk of HCC cancer?
Response: Specifically, we chose the STAT4 rs7574865 polymorphism due to its prior association with various autoimmune diseases and its role in the JAK-STAT signaling pathway, which has been implicated in liver fibrosis and hepatocellular carcinoma (HCC). This polymorphism was selected based on existing literature that hints at its possible role in the modulation of immune responses, inflammation, and consequently, HCC risk. We improved the introduction by adding the following statements: “This SNP has been previously associated with autoimmune diseases such as autoimmune thyroid disease, ulcerative colitis and malignancies, such as HCC [15,16]. STAT4 expression has also been associated with numerous malignancies such as ovarian and lung cancer [17,18].” (page 2 line 27). We also include information regarding the mechanisms in this section with “It has been theorized that reduced STAT4 expression, as induced by the SNP, could potentially weaken the anti-tumor response by affecting STAT4-modulation of interleukin 12 in NK cells” (page 2, line 30). As well as “Building on this mechanistic understanding, numerous studies have been conducted to assess its clinical relevance, confirming a notable association between this particular SNP in STAT4 and HCC in different populations. Jiang et al. found an association between STAT4 and HCC among patients with HBV related HCC in a genome-wide association study (GWAS) performed in China [14]. While other studies such as the replication study by Chen et al. failed to repeat the same findings [20].” (page 2 line 29).
- The discussion part is rather simple, mainly repeating the results, but generally lacking the real discussion of obtained results with previously published studies of association of selected polymorphisms with either HCC cancer or other oncological diseases. Mechanisms for the discrepancy data should be clarified.
Response: We have significantly expanded the discussion including comments and comparison with previous studies and providing potential mechanistic explanations for the difference found. (page 9 line 1, page 9 line 16, page 9 line 32).
- In materials and methods, no data about the selection of rs mutation such as Assay ID,… so readers and researchers can reproduce data using the same assay rs.
Response: We now added the following information on the primer-probes to the revised manuscript “(cat nr. 4351376). The context sequence of the probes is TATGAAAAGTTGGTGACCAAAATGT[G/T]AATAGTGGTTATCTTATTTCAGTGG, with the variation G versus T as indicated” (page 3 line 43.).
- No additional post-tests have been performed to confirm the genotyping data either on gene (mRNA) or protein level.
Response: Our study aimed at understanding the association between STAT4 and HCC in a specific population, but not to evaluate the impact of STAT4 SNP on the function of the protein. Although this is interesting it goes far beyond the point of our manuscript (and it would be a study by itself). Other studies have addressed a direct correlation between STAT4 rs7574865 and STAT4 expression (reference 14, PMID: 23242368)

Reviewer 2 Report
The study investigated the association between STAT4 rs7574865 G>T SNP and hepatocellular carcinoma (HCC) in Latin American and European populations (n=1,060). No statistically significant correlation was found between the SNP and HCC risk in either cohort, or subgroup analyses based on underlying liver disease also yielded no significant associations. Ethnic background and self-reported heritage were considered, emphasizing the need for cautious extrapolation of risk profiles across populations. The manuscript is well-organized and presents important insights into the role of the STAT4 rs7574865 G>T SNP in HCC risk. However, I suggest the following enhancements to strengthen the manuscript.
Comments for authors:
Title:
The title accurately reflects the content of the study and is concise and informative.
Abstract:
1. The abstract clearly states the research objective, which is to investigate the association between the STAT4 rs7574865 genetic variant and HCC risk in Latin American and European populations.
2. The abstract could benefit from further clarity in terms of the methodology. The term "samples" could be replaced with "DNA samples," and more details about the genotyping technique could be provided.
3. While the abstract mentions the lack of association between the risk allele and HCC, it does not provide specific data (e.g., odds ratios, p-values) to support this claim. Including such information would enhance the abstract's strength without affecting the limitations of the abstract length
Introduction:
1. The transition from discussing the SNP's mechanistic hypothesis to the mention of previous studies in Asian and Caucasian populations feels abrupt. A smoother transition and contextualization of these studies could enhance the flow.
2. While the section mentions previous studies on the STAT4 SNP, it could benefit from a brief summary of their findings and methodologies to provide readers with a clear understanding of the existing knowledge gap.
3. The section outlines the study's objective, but it could explicitly state the hypotheses being tested regarding the association between STAT4 rs7574865 SNP and HCC in the Latin American population.
Methods:
1. Expand on the criteria used for continuous participation in the ESCALON study, highlighting its relevance to the study objectives and potential implications for participant selection bias.
2. Provide more detailed explanations for the American Association for the Study of Liver Disease's standards for HCC diagnosis and the criteria for classifying patients with alcohol-related liver disease and MASLD
3. Briefly explain the principles of TaqMan SNP genotyping and how it detects SNP variations in DNA samples.
4. Elaborate on why JASP software was chosen for the statistical analysis, highlighting its suitability for the study's requirements
5. Clarify whether the informed consent process was reviewed and approved by the relevant ethics committees and how participants were informed about the study.
Results:
1. Abbreviations like "CI" and "OR" are used without immediate explanation, which could be confusing for readers not familiar with these terms.
2. The figure legends should provide more detailed explanations of the contents of each figure, especially for Figures 2 and 3, which appear to involve comparisons between different patient groups and genetic factors
3. Label the x and y axes of Figures 2 and 3 to indicate the parameters being compared, and provide a brief title for each figure to clarify the content.
Discussion:
1. The section could benefit from incorporating specific citations to support the statements made, especially when discussing the effects of STAT4 SNP in other populations.
2. To avoid potential confusion, provide a brief explanation of why self-reported heritage background was included as a factor in the study, even without genetic analysis.
3. If possible, provide a more detailed comparison with previous studies that reported an association between the STAT4 SNP and HCC, highlighting potential differences in patient populations, ethnic backgrounds, and underlying causes of liver disease.
4. Emphasize the implications of the study's findings for clinical practice and future research directions. Discuss how these findings contribute to the broader understanding of HCC risk and potential clinical applications
Conclusions:
1. There is no conclusions section.
Author Response
"Please see the attachment"

Round 2
Reviewer 1 Report
Regarding point (4), post-tests, the authors needs to show in Discussion the future prespective regarding track of the protein or other molecular targets to confirm the effect.
Author Response
"Please see the attachment."
